# The Internet of Drones: Requirements, Taxonomy, Recent Advances, and Challenges of Research Trends

**DOI:** 10.3390/s21175718

**Published:** 2021-08-25

**Authors:** Abdelzahir Abdelmaboud

**Affiliations:** Department of Information Systems, King Khalid University, Muhayel Aseer 61913, Saudi Arabia; aelnour@kku.edu.sa

**Keywords:** Internet of drones, communication, security, privacy

## Abstract

The use of unmanned aerial vehicles or drones are a valuable technique in coping with issues related to life in the general public’s daily routines. Given the growing number of drones in low-altitude airspace, linking drones to form the Internet of drones (IoD) is a highly desirable trend to improve the safety as well as the quality of flight. However, there remain security, privacy, and communication issues related to IoD. In this paper, we discuss the key requirements of security, privacy, and communication and we present a taxonomy of IoD based on the most relevant considerations. Furthermore, we present the most commonly used commercial case studies and address the latest advancements and solutions proposed for the IoD environments. Lastly, we discuss the challenges and future research directions of IoD.

## 1. Introduction

The Internet of drones (IoD) can be described as an infrastructure designed to provide control and access over the Internet between drones and users. In reality, drones are rapidly becoming readily available commodity items, allowing any user to fly different missions in controlled airspace using multiple drones. Although technology helps mass-produce the onboard components of unmanned aerial vehicles (UAVs), including processors, sensors, storage and battery life, the performance limitations of these components impede and reduce expectations. IoD offers drones coupling vehicle as well as cloud mobility functions to allow remote drone access and control, as well as seamlessly scalable offloading and capabilities of remote cloud storage [1,2]. Figure 1 illustrates the IoD environment that includes base stations, signal link, and cloud environments.

The key advantage of a UAV with fixed wings compared to a UAV with rotary wings is that a less complex repair and maintenance process is required by simple structure, thereby giving the consumer more operating time at a reduced cost. The simple structure offers a high-speed aircraft that can have a longer flight length and can cover more ground. UAVs may make use of non-power supply techniques to make gliding more efficient. It is also worth noting that fixed-wing airplanes can carry a greater payload for longer distances when flying with less power giving them the capability to carry a combination of bigger more advanced sensors with a pair of complementary sensors [3].

Until recently, UAVs were operated individually, but today a higher number of coordinated drones may work together to accomplish complex missions. In these circumstances, drone communication is absolutely essential. In other words, it is vital for users to fully comprehend UAV communication systems. One additional kind of wireless channel and network protocol is utilized in drone communications, but on the other hand, several distinct types of wireless routes and network protocols are applied in drone communications. For this reason, the network design for UAVs is determined by their application. As a basic example, researchers have discovered that a point-to-point line-of-sight link between a drone and a gadget may maintain continuous data transmission even when transmission is extended. Drones that use satellite communications to talk to each other for surveillance, when employed for safety defense, or more broad outreach activities, satellite communication is a better option for drones. Alternatively, cellular communications systems are more commonly used in civic and personal applications. For example, indoor communication, in particular for the mesh network and WSN, P2P protocols such as Bluetooth have shown to be more efficient. When applied to drones, working with a multi-layered network can be a difficult and challenging procedure [4]. Examples of important problems are outlined below.

A remote hijacking of the drones could be achieved by leveraging the vulnerability in the software of the UAVs that act as a sophisticated tool for military purposes. Global positioning system (GPS) signals are under the influence of malware programs on drones that can be controlled by malicious users for malicious objectives (attackers). By doing this, unreasonable attacks, such as dropping bombs, could be committed by the attacker, endangering lives. The control signal is a significant feature of IoD environments due to the different communications among entities and should not be disclosed or exposed in any circumstance. There is a need for robust security measures to avert harm from security attacks. Moreover, to facilitate personal and business drones for independent flight, a certain type of authentication and key exchange protocols are required between the two entities in the sky. Both the entities then create a symmetric security key for future data transmission [4,5,6].

The rise in the popularity of drones has increased the frequency of cyber assaults against UAV systems during the last decade. An adversary will target the radio connections of UAV systems in order to hinder the systems’ ability to communicate with user equipment. This includes data required by the user’s cellular equipment that control signals and GPS signals. This is an example of an interception of information giving enemies the ability to steal data that drones transmit and request, as well as foregoing direct control of the drones through control signals. To guarantee the security of the wireless communication channels, guaranteeing that both control and data signals are sent has become a critical element of the overall UAV system [7]. This review focuses on the recent advancements in IoD development. The key contributions to this paper, which are presented from Section 3, Section 4, Section 5, Section 6, Section 7 and Section 8, are to:Investigate the main requirements of IoD (Section 3).Provide the main parameters-based thematic taxonomy (Section 4).Discuss the most commonly used commercial case studies of IoD (Section 5).Investigate the progress recently described in the literature (Section 6).Discuss the IoD open challenges and prospective study directions (Section 7).

## 2. Related Works

In order to examine the possible solutions involving the use of the Internet of drones, several survey studies have been published in the literature. The specific surveys are briefly summarized in this section and given in Table 1, which also covers the substance of this article.

Boccadoro et al. [8] conducted an overview which focused on the consequences of widespread drone adoption for the economy and society. They covered many aspects of the IoD, including a comprehensive analysis of privacy and security issues. Additionally, they explored on what are considered to be the primary research problems and projects that are of high interest to the IoD for prospective future developments.

Rejeb et al. [9] investigated drones to discover their skills, achievements, and challenges in logistics for humanitarian use. They provided an in-depth overview of the capabilities, challenges, and results associated with humanitarian drones in logistical operations, management, and governance. Furthermore, they examined possible humanitarian uses for drones and established a road map for more extensive research on the issue. Yahuza et al. [10] conducted a review to seek and evaluate the factors that impact the IoD network security and privacy. They examined the variety of drone danger levels and security and privacy vulnerabilities. Additionally, they analyzed the recently developed IoD security controls. In addition, they provided the methods for determining the performance, such as assessment and measurement procedures, that are used by the approaches.

Ayamga et al. [11] reviewed the advancements in agricultural, military, and medical drones to showcase the strengths, weaknesses, opportunities, and threats of the current technologies. In addition, the study suggested a need for further research investigation to better integrate drones into current transportation and supply chain infrastructure, such as increasing payload and flight length, and addressing the unique aspects of a country’s culture in which drones are easily accepted and embraced.

Merkert and Bushell [12] conducted a review of systematic literature related to historical concerns such as privacy, acceptance, and protection that have been gradually replaced by operational considerations, including contact with and impacts on other users of airspace, in order to recognize critical issues and research gaps. The study demonstrates that unrestricted use of drones can cause problems for other users of the airspace, such as airports and emergency services. In addition, analysis of current regulatory approaches illustrates the need for more policy and management response to both handle rapid and efficient growth in drone usage and promote innovation, with systems of the airspace management for all cases of drone use being one promising strategic response.

Zaidi et al. [13] reviewed the state of the art with an emphasis on the Internet of flying things (IoFT). They proposed taxonomy literature, including description, classification, and comparative analysis of various IoFT works. In addition, the study presents an applications problem for IoFT and future perspectives. Moreover, the survey provides scientific researchers with basic concepts and a full overview of recent IoFT studies.

Yaacoub et al. [14] conducted an overview and analysis of the use of drones or unmanned aerial vehicles (UAV) in multiple domains such as civilian and military, terrorism, and for other various purposes. They describe how a simulated attack on a given drone was carried out by the authors following the hacking period. The study analysis will significantly assist ethical hackers in recognizing the current vulnerabilities of UAVs in both military and civilian realms. In addition, their proposed future study will enable implementation of new techniques and technologies for the identification and defense of enhanced UAV attacks.

Al-Turjman et al. [15] highlighted an overview of drone applications, emergency networks, and surveillance monitoring in software-defined networking-enabled drone-base stations (DBS). Furthermore, they reviewed the performance assessment techniques and related aspects of cybersecurity applications. In addition, the study presents the necessity to advance in IoT-enabled spaces by developing an innovative and multi-faceted drone performance assessment framework with primary concerns, meeting user-defined requirements, and providing safe and reliable services.

Alsamhi et al. [16] presented a survey of the possible collaborative drone and IoT techniques and technologies recently proposed to increase the intelligence of smart cities. Their study focused on data collection, privacy and protection, public safety, disaster management, energy use and quality of life in smart cities, and enhancing smart cities’ smartness.

Fotouhi et al. [7] presented an overview of advancements supporting compatibility of UAVs into mobile networks. Their study focuses on discovering what types of consumer UAVs are available to purchase off-the-shelf, and what other people and organizations have addressed when it comes to standardization for serving aerial users with the existing terrestrial base stations. In addition, the study investigates the potential issues and solutions that are being addressed by standardization bodies for aerial users with respect to interference.

Wazid et al. [4] discussed certain security issues and IoD environment specifications. They also discuss a taxonomy of different security protocols in the IoD setting. The study emphasizes the analysis of some of the user authentication systems recently proposed for IoD communication. In addition, the study defines the strengths and limitations of user authentication through a systematic comparative analysis of existing systems with IoD contact systems.

Bagloee et al. [17] discussed an overview of literature related to computer ethics and safety including vehicles’ connection and infrastructure of future growth of autonomous vehicles (AV). The study illustrates that a significant information gap exists in AV technology with regards to route activities. In addition, the study shows a tremendous opportunity to incorporate a routing system that offers an effective and intelligent connected vehicle technology. Gharibi et al. [1] presented a conceptual model of how such an architecture can be designed and defined the features that should be implemented by an IoD framework. They investigated and extracted key concepts of current large-scale networks such as cellular network, air traffic control network, and the Internet to link with drone traffic management architecture.

## 3. Requirements of Internet of Drones (IoD)

The expected increase in the use of drones in a variety of applications could expose operators to a whole new array of risks, particularly damages to third parties and liability. Several of the requirements for potential drones are presented and classified in the following sub-sections. Figure 2 shows the key requirements of IoD.

### 3.1. Communication Requirements

The impact of IoD communication vulnerabilities is receiving increasing attention from researchers. Many remote locations would be hard to reach were it not for the usage of drones. As a result, drones are commonly employed for important tasks such as rescuing victims, providing surveillance, transportation, and helping with conserving and protection of the environment [18]. Therefore, critical communication requirements to support the different drone applications are discussed as follows.

#### 3.1.1. Seamless Coverage

Hot-spot coverage (stages, tourist areas, and industrial areas) is appropriate for aerial entertainment. Widespread coverage in suburban, urban, and rural areas is required for the inspection and logistics of power or base stations. In the future, seamless drone coverage will become more essential for network planning. Unlike conventional network coverage serving mostly land users, enhanced sky coverage is required to support drone users flying at different heights.

Coverage of up to 10 m altitude is appropriate for plant protection (e.g., spraying of agricultural chemicals). Coverage of up to 50 to 100 m altitude is required for power line inspection. Coverage of up to 200 to 300 m altitude is sufficient for mapping of agricultural lands, while coverage of the upper air pipeline of up to 300 to 3000 m altitude may be needed. It is difficult for networks to serve this large spectrum of coverage scenarios at varying altitudes [19].

#### 3.1.2. Real-Time and Remote Communication

Real-time and remote communication capabilities permit remote controllers to issue a time-based command and control instructions on the basis of the drone flight status report in real-time, such as space co-orders and equipment status. Real-time and remote controls are primarily used in the monitoring of flight conditions, drone task, and equipment and emergency control. The latency and rate of certain data requirements should be met to allow remote control for drones. The downlink (from the base station to drone) data rates in many application scenarios are about 300–600 kbps, and the existing 4G+ networks will fulfill this requirement. For potential implementations, such as remote real-time operations, the latency criterion is strict to guarantee the precision and experience of service [19].

#### 3.1.3. Transmission of HD Image/Video

Networks should be able to provide a high uplink (from the drone to the base station) data rate for drones to permit the transmission of HD image/video. The data rate required is calculated mainly by picture/video size and quality.

In the future, the demand for higher resolution images/videos in vertical industries needing 4K/8K HD video support would need a higher Gbps-level data rate. The 5G networks are well equipped to support such services with a data rate requirement of multi-Gbps. Transmission of HD image/video will dramatically extend the drones’ application scenarios including energy and power line inspection, agricultural exploration, control and rescue, entertainment and monitoring. With high transmission rates, drones linked to networks are able to transmit HD images/videos to enable the immersive experience of augmented reality and virtual reality [19].

#### 3.1.4. Drone Identification and Regulation

Mobile networks may help identify and control drones by supporting drone registration, tracking, provision, and coordination [19,20].

Registration: Identifying but standardizing the drone equipment number, serial number, and flight control serial number helps track the whole process orderly from initial drone production to in-use. Through standardizing registration of drone users, owners and mobile networks, drone users and owners can be legally monitored.Monitoring: Drone connections and data communications can be detected and monitored through mobile networks. Drone implementations can be completely tracked in real-time with additional regulatory protocols.Forecast: Flight situations can be dynamically evaluated and early warning of possible risks can be achieved by tracking drone positions and monitoring the flight traffic and path.Coordination: Knowledge exchange between industries and different companies can be carried out by approved oversight of all vertical industries involved.

#### 3.1.5. Positioning of High-Precision

For numerous drone applications, positioning is critical. In several drone applications, vertical positioning also is important, in addition to traditional positioning on the horizontal plane. The requirement for positioning accuracy will increase from tens of meters to sub-meters with the drone applications’ development. Fifty meters positioning precision is adequate for regular monitoring activities. Applications such as agricultural land mapping and automated loading involve high precision positioning at the sub-meter [19].

### 3.2. Security Requirements

Researchers have created several security and privacy approaches to secure the Internet of drones (IoD) network to protect the location of unmanned aerial vehicles (UAVs) and the privacy and security problems that come with using the IoD network. Due to these localization errors, drone positioning was previously unreliable, which had disastrous implications for the whole IoD network. Another critical aim of the IoD network is to increase the level of security and privacy to a point where it cannot be compromised [21]. Therefore, the main requirements of security and privacy of the IoD network include authenticity, confidentiality, availability, integrity, and non-repudiation [4,11,22]:

#### 3.2.1. Authenticity

Authentication is required for sensing devices, users, and portal nodes before access to a limited resource is enabled or essential information is disclosed [23]. In addition, for communicating entities to have mutual authentication, two of the communicating entities have to be a monitoring drone and a ground-control station. To guarantee full forward secrecy, it is essential to employ a secure key exchange using a method that produces session keys that are impossible to recover [24].

#### 3.2.2. Confidentiality

Confidentiality or privacy of the wireless communication channel protects from the unauthorized disclosure of information [24]. Another significant barrier to IoD implementation is making data available, and controlling access to that data (data confidentiality). For instance, when a group of drones collects road traffic data from several places, there is a continuing problem in sharing this data safely and effectively [20].

#### 3.2.3. Availability

Registered users should also be granted access to appropriate network services in conjunction with system denial-of-service attacks. Both the mechanism and the system are capable of recognizing if a drone is engaged in combat and keeping track of the battle limit, which governs whether the flight management system can pick up on a malfunctioning drone and determine if the availability criteria are compromised [10].

#### 3.2.4. Integrity

Integrity is essential to guarantee the trustworthiness of the information (for example, that it has not been altered in transit, and the source of the information is genuine) [24].

#### 3.2.5. Non-Repudiation

The goal is to ensure that a criminal organization does not conceal its actions. In fact, when there are several parties conducting an action, one of the necessary security measures is to make sure that the action cannot be rejected without the others’ knowledge [24].

## 4. Taxonomy of IoD

This section presents a thematic taxonomy for the Internet of drones that consists of the elements of IoD, notably architecture, middleware, data fusion and sharing, security and various applications, as shown in Figure 3.

### 4.1. Architecture

IoD architecture is divided into two major elements: Architecture components and communication protocols. IoD architecture plays a vital role in controlling and administrating unmanned aerial vehicles to perform their operation more efficiently [25].

#### 4.1.1. Architecture Elements (Component)

There are numbers of the Internet of things components present in the architecture of IoD. The role and operations of these components depend on interaction with the IoD architecture. Furthermore, stability, data acquisition, and communication methods of IoD are determined by its architecture components [26]. In addition, IoD architecture components are intended to perform operations mainly related to deciding and controlling the drones and ensuring that the data reach the correct destination from the source nodes [1,27]. Lastly, in the development of IoD architecture, we have to consider resources such as airspace, intersections, and nodes [1].

#### 4.1.2. Communication Protocols

IoD enables multiple communication protocols to support and transfer data between the nodes. Authors propose a drone planner which assists in two communication protocols, namely the MAVLink protocol and ROSLink protocol. The MAVLink is a lightweight message marshaling protocol and ROSLink is integrated with the robot operating system which provides robots into the IoT [28]. The efficiency of the communication protocol is decided by the node movements and behavior in the network [29]. Furthermore, the communication protocol for IoD has to accelerate in routing competencies where data communication between the source node to target nodes is crucial [30]. Due to the above reasons, selection of the communication protocol for IoD should be analyzed thoroughly.

### 4.2. Middlewares

In a connected IoT devices environment, the middleware layer plays an essential role functioning as a mediator between various nodes and applications [31]. Moreover, the middleware layer ensures abstraction between the various interfaces of the IoD, namely programming language, operating system, networks, and architecture [32]. Two major middlewares, service-based and cloud-based are discussed in the sub-section.

#### 4.2.1. Service-Based

Service-based middleware facilitates in offering network access, local message delivery, caching and name resolution to the IoD architecture [33]. Service-based middleware can achieve robust connectivity for the entire IoD architecture. Furthermore, it can integrate well with other network layers to provide efficient collaboration, performance, and is adaptive to the architecture. Authors propose service-based middleware that enables effective data transmission and extended connectivity time between the network communications [34]. In IoD architecture, simultaneous communication between the ground network and drone is a fundamental principle. The authors propose concurrent communication between ground ad hoc networks and multi-UAV network for on-time communication [1].

#### 4.2.2. Cloud-Based

Cloud-based middleware for IoD service facilitates diverse applications to integrate their operations into the network architecture. The most commonly used robotics application middleware is the robot operating system (ROS) which utilizes the benefits of its elastic resources by offloading computation and processing to the cloud for efficient resource usage. This process is named cloud robotics [1]. The cloud-based middleware provides reliable communication between the ground network and UAV [35]. The cloud-based middleware delivers a very rapid response to the requested service. It further benefits the IoD to actively communicate with the other networks.

### 4.3. Data Fusion and Sharing

Currently, multiple drones are connected to perform various operations simultaneously. Incorporating data fusion and sharing for IoD allows the processing and merging of several data sources to generate correct information for decision making. Moreover, data fusion based algorithms for multi-UAV further support more efficient performance in the smart environment. In this sub-section, we outline the three different types of data fusion and sharing for the IoD, namely distributed, centralized, and cloud-based.

#### 4.3.1. Distributed

The distributed data fusion and sharing are also called meta-architecture. This decentralizes the data into local interaction and no components are important to any other operation and results in several benefits for the IoD operation, namely scalability, being fault-tolerant, interoperability, and ease of redesign [36]. These benefits help reconfigure or remove the particular unavailable or disconnected drone from the network. Moreover, the distributed environment is more convenient to make a decision independently and in the case of emergency can work as collaborative sharing.

#### 4.3.2. Centralized

The centralized data fusion and sharing mechanism needs high-power communication equipment to perform the operation smoothly. Furthermore, it shares all the information through the fusion center with other devices in the network and leads to provision of more accurate information about the operation [37]. At the early stage of deployment and operation for the shorter distance, centralized sharing performs more efficiently. However, when it comes to longer distances and later part of the operations, it will be much more complicated to reposition or control the device [38].

#### 4.3.3. Cloud-Based

Cloud-based data fusion and sharing are completely controlled by cloud interfaces that consolidate various services to support the client to make an effective decision. Furthermore, services such as analytics and intelligent operation can be performed in the architecture. The developed framework to integrate various sources to operate traffic management of smart cities applications use cloud-based data fusion. In addition, these frameworks offer services to safety, collision avoidance, and risk-aware navigation [39].

### 4.4. Security

IoD transport and convey by more sensitive and confidential information about the operation. Correspondingly, unauthorized access or control to drones leads to the destruction of any services or physical attack. In this sub-section, an overview of four different areas in security is discussed.

#### 4.4.1. Authentication

Authentication is one of the major threats to any drone, whereby information disclosure results in leakage of identity, location, flying routes, and other meta-data information about the device. To improve the authentication, the authors propose a fast modular arithmetic operation for the drone that helps control the signature key generation and computing capability of any UAV resources. In addition, preserving authentication further secures the authentication performance for UAV-to-UAV in real-time [2]. Moreover, authentication should be given higher priority when it comes to accessing information in real-time, otherwise the attacker can change the original message while operations are performing in real-time. Most of the communication channels are protected by encryption algorithms, which help protect the unauthorized authentication of the IoD [40].

#### 4.4.2. Privacy

The privacy issues of the IoD arise from three major areas: Sensors, communications, and multi-UAV [41]. Eaves-dropping and keylogging are the popular methods in data interception which is aimed to affect the privacy of the flowing information between the drone and communication center [42]. Furthermore, the IoD utilizes the open-access communication environment which is vulnerable to many security and privacy issues such as message integrity, and disclosure of the UAV identity [2]. The recent advancement in drone usage by civilian and business entities further increases the possible harm to the right of the individual or organization to privacy [43].

#### 4.4.3. Intrusion Detection

An intrusion detection system is classified into two main categories, namely host and network-based. The authors propose a hybrid method for the UAV network intrusion detection system, which comprises spectral traffic analysis and a robust controller to observe the abnormal behavior in the UAV environment. Moreover, it performs a statistical signature to explore the threat. Furthermore, the wireless network faces several threats, namely overload, flash crowds, worms, port scans, jamming attacks, etc. [44].

To handle the above-mentioned modern threats, integrating deep learning and big data technologies for intrusion detection will provide a more efficient method to identify the threats [45].

#### 4.4.4. Availability

The specialty of the IoD is to have high availability of the data flows in real-time to the network. The IoD streams the high rate of data in the emergency response operation [46]. Moreover, in an emergency, the rivals may alter the incoming drone data or stop the availability of the data to threaten the current situation. This would affect the evacuation operation in the area [47]. Apart from this, these atomic attacks, namely battery exhaustion attack, fuzzing attack, and physical component warning suppression would be performed to disrupt the availability of drone data [48]. Lastly, any modern security system should protect the collected drone data to be available in the network without interruption for further analysis.

### 4.5. Applications

In this section, we discuss the most frequently used IoD applications, namely smart agriculture, mining, construction, emergency and delivery services, and film and TV. Table 2 illustrates the summary of drone applications and its modern usage areas.

#### 4.5.1. Smart Agriculture

IoD based smart agriculture is being pursued to satisfy the growing demand emerging from an environmental change with an ever-increasing population and limitation to natural resources. In general, due to over-reliance on machinery, many farmers have left behind farming jobs leading to a shortage of workers for agriculture industries.

Innovations such as IoD, cloud, and IoT will help them monitor their fields in real-time to make an efficient decision. In particular, adopting the IoD and wireless sensor network for smart agriculture will add more value and efficiency in monitoring and growing crops for better yields. Furthermore, IoD eases the process by aerial monitoring and smart spraying in crop affected areas [49]. In the era of precision agriculture, remote sensing with IoD and GPS technologies further helps increase the various types of detection in the farming sector, such as drought stress, nutrient status, and weed detection [50].

#### 4.5.2. Mining

The usage of the IoD in mining applications protects human safety and reduces the cost of mining operations. Moreover, the disruptive technologies help in indoor and outdoor visual inspections, pit planning management, stockpile surveying in the mining industries. The collected IoD data can be further processed to make efficient decisions in mining industries. The authors propose that the framework of UAV applications in mining areas is divided into three major categories. Firstly, basic data from digital cameras, spectral, lidar, thermal infrared, and gas sensors. Secondly, monitoring objects such as dump, surface subsidence, coal gangue heap, open mining pit, and industrial site. Thirdly, different applications in mining areas [51].

#### 4.5.3. Construction

IoD makes it easier to access the visual data of the construction site to the client’s view simultaneously. Furthermore, connected sensors, GPS, and the high-quality camera provide various details to different stakeholders in the project. With sophisticated technologies, the project manager has real-time data access to the site to monitor and avoid waste in the construction process.

In addition, worker smart helmets in the construction site provide data about the safety of an individual person to the drone. However, usage of IoD for the construction industry is quite new compared to other application areas which need clear regulatory approval and availability of experts [52].

**Table 2 sensors-21-05718-t002:** Summary of drone applications and their modern usage areas.

Applications	Modern Usage Areas	Reference
Smart Agriculture	Soil and Field Analysis Crop Monitoring Irrigation Pathogen detection Drought stress Weed detection Nutrient status Yield prediction	https://www.microdrones.com/en/content/drones-and-precision-agriculture-the-future-of-farming/ (accessed on 10 December 2020). [47,49]
Mining	Field measurements Water sampling Stockpile management Site description Mine or quarry monitoring Sediment flow Hazard identification	https://wingtra.com/drone-mapping-applications/mining-and-aggregates/ (accessed on 10 December 2020). [51,53]
Construction	Construction Safety Autonomous operation Planning and analysis process Site Inspections	[52]
Emergency and Delivery Services	Rescue Operations Blood and medicine Transportation Patient Monitoring Disaster relief and Management Find Missing Persons Public safety	[53,54,55]
Film and TV	Scouting and planning Location managers Logistics and safety Multiple sensors shooting Visual effects	https://variety.com/2019/artisans/production/film-and-tv-shoots-drones-1203223079/ (accessed on 11 December 2020). [32,56]

#### 4.5.4. Emergency and Delivery Services

IoD plays a significant role in forest search and emergency medicine delivery to remote areas. Furthermore, it is used for transporting blood, for disaster relief, missing individuals finding lost hiker in the hill station, and many more areas of emergency services. Tech giants, such as Google and Amazon use the sky to provide medical care and food delivery. IoD is equipped with a unique signature to track individual missions to verify the place to which they traveled for disaster assistance. This helps coordinate the search and rescue more efficiently. IoD is even used in the assistance of underwater search and rescue operations. However, the greatest challenge relates to the bandwidth of IoD devices that need to assist with intelligent algorithms [53].

The use of drones for emergency response services, especially in medical situations, opens up new possibilities for life-saving measures. Using drones to provide “eyes” on a dangerous situation or to deliver medical supplies to stranded patients may improve emergency response physicians’ ability to provide treatment in dangerous situations. IoD offers many emergency response services that impact daily life, discussed as follows.

Disaster management: Sensor data are a major problem in a broad geographical area. In the case of satellite remote sensing, the identification of different parameters such as forest vegetation, the tracking of the bottom of rivers, and the desert is very difficult. For these situations, most devices are specifically designed and have very low time sensitivity. In addition, environmental instability such as large clouds, cyclones, and volcanic ash can cause disturbances. A certain situation can occur when acquiring large-scale sensor data.

As is the case in most situations, the sensors are located in fixed positions and the network architecture is not suitable for rapid monitoring and management. In addition, the sensor nodes themselves are susceptible to failure and low transmission capacity. Thus, a dynamic node collection is highly important, which links the ground station and data processing sites by utilizing the utility of the drone ecosystem Internet. Furthermore, the drone functions as a highly complex data mule and sensing nodes, and sometimes the nodes are disconnected from the network. That is, the intrinsic phenomenon of flying nodes in dissipated networks and ad-hoc flying networks. Therefore, clearly, the nodes interconnect in an opportunistic way [54].

Public safety: Whether a delivery site or a specific restaurant, and being more popular with COVID-19, we can see that replacing personal delivery with a drone can boost protection (no chance of transmission) but there are other possible benefits of drone usage. Fewer cars will minimize CO_2_ emissions and alleviate congested streets. Drones may reduce delivery times in non-urban areas or enable delivery to previously unreachable areas. Cost savings will also potentially occur, the average cost of owning and using a vehicle appears to be greater than a drone. Despite the labor pool, many restaurants and supermarkets had trouble filling open vacancies prior to COVID-19. The use of drones can allow the redeployment of their labor by these establishments [55].

#### 4.5.5. Films and TV

Films and TV production are going through revolutionary changes from human intervention to automation. Algorithms, technologies, and expert knowledge systems provide more efficiency to produce a high quality of creative content into cinematography industries. In the existing approach, labor workers are heavily used to position, and target the camera position for adjusting the camera motion. These problems are easily addressed by a drone-based intelligent shooting system, which further integrates with machine learning and computer vision to assist in more convenient automation.

IoD contains more than one high clarity camera to track the visual object for the production of films and TV. Moreover, IoD provides high precision and coordination to capture the shot for cinematography applications. However, the standardization of drones for shot types and camera trajectories need to be addressed [56]. Films and TV had comprehensively addressed the various existing elements in IoD, in particular, architecture, middleware, data fusion and sharing, security, and various applications.

## 5. Commercial Case Studies

A broad variety of Internet of drone applications in the fields of mining and industrial automation services are being opened up by drones acting as sensor devices. Drones are now being designed for a variety of business applications that are an increasing using IoD. The following section discusses the most key commercial case studies of IoD, summarized in Table 3.

### 5.1. Matternet

Matternet is the leading technology company for independent urban drone logistics systems, offering the technology platform as a service for healthcare and logistics organizations for the transport of medical supplies through dispersed healthcare networks. Matternet has successfully deployed drone networks all over the world and has been licensed by the Swiss Federal Office of Civil Aviation (FOCA) and the United States for regulatory purposes. The Federal Aviation Administration (FAA) carries out fully autonomous operations beyond the line of sight and flights over people to carry essential, important medical supplies. To date, over 10,000 commercial revenue drone flights have been facilitated by Matternet technology [57,58].

In order for Matternet to operate drones in Zurich, local regulators required a flight permit. Drone reliability and safety were crucial regulatory requirements. Regulators required that human operators access the status and location of the drone in real-time and had the ability to manually interact with and monitor the drone, if necessary. Moreover, Matternet had to incorporate data on air traffic control in real-time since helicopters frequently landed near hospitals. The built-in Matternet solution includes self-employed drones and landing stations, as well as a logistics cloud platform. The operations platform is in charge of flight planning and drone surveillance in real-time. A flight director can track the flight status and send orders to drones to come home or wait at certain places [59].

### 5.2. ARIA

Autonomous roadless intelligent array (ARIA) insights use artificial intelligence (AI) and drones to eliminate people from unpredictable circumstances. Smarter collection of data and machine learning allow decision-makers to solve a problem or conduct a task rapidly and efficiently, while ensuring that human lives are not placed at risk. Decision makers no longer have to spend hours recording and watching a video with sophisticated analytics. Instead, the robots from Aria insights can recognise information that is relevant, sound an alarm when new data are detected, and eventually link all the data to a digital 3D map [60].

The new machine learning technologies of Aria insights require consumers in the public safety, oil and gas, and other commercial sectors to gather information and then transform it into actionable plans. Aria’s drones can conduct fully autonomous missions fitted by AI capabilities to make it easier to recognize characteristics of interest in multiple use cases. For example, drones may help maintain enclosed spaces, such as oil tankers or pipelines, allowing operators to locate areas that need to be repaired without human exploration into potentially unsafe working conditions. Aria insights drones can also track the progress of a natural disaster and inform first responders about increased activity in a given area, as well as warn event organizers when sections of the crowd may require more supervision [60].

### 5.3. DroneSmartX

DroneSmartX, a UAV service provider, has created the UAV SmartHub, a system that allows users to connect to smart sensors on the ground and send data directly to the cloud platform. According to the company, headquartered at the University of Central Florida’s business center in Kissimmee, Fla., the UAV SmartHub is connected to several types of commercial-grade drones. DroneSmartX claims that the device integrates UAV as a service with the Internet of things to make drones smarter [61].

The platform gathers data from sources such as passive and active radio frequency identification (RFID), Bluetooth, GPS, and other sensors; processes the data; and delivers it back to the user through the cloud platform in real-time. Ideal applications include agriculture, energy, manufacturing, oil and gas, and inventory of warehouses. According to DroneSmartX, the SmartHub will add tags to thousands of items for inventory checks in minutes [61].

### 5.4. H3 Dynamics

H3 dynamics is developing the future of business drone services through the combination of machine learning, remote tele-robot, and off-grid power to ensure the deployment and management of condition control, surveillance, and security solutions from anywhere. H3 dynamics provide huge volumes of smart, professional-validated inspection reports into complex images, videos, and other sensor data, a cloud-based service for owners, airlines, contractors, and financial companies [62].

H3 dynamics have developed a station named “DBX” that can easily link all its functions with third-party networks, including particular security and communications networks, using an open application program interface (API) architecture. The DBX can respond in any mobile or fixed sensor to geo-located signals, such as an intelligent camera in a CCTV network, mobile phone, fire sensor or radar coordinate after an earthquake or movement of some sort. To facilitate the fight against COVID-19, Rooftop DBX drone facilities are being re-used as rooftop drone distribution and delivery networks for test kits, blood samples, and provision of supplies. H3 dynamics provide services to many sectors such as real estate, mining, and oil and gas [62].

## 6. Recent Advancements

This section explores recent advancements in IoD research activities (models, scheme frameworks, protocols, mechanisms, methods, architectures, and algorithms). Table 4 lists the ideas that have been put forward and they are addressed below.

Almulhem [63] proposed a model for a security threat, known as threat trees, to evaluate and enumerate threats to the architecture of the IoD. The suggested threat tree is intended to provide a holistic view of threats affecting an IoD scheme. However, the threat tree that has been proposed does not necessarily include all potential threats.

Zhang et al. [64] suggested a scheme to enable drones and users to authenticate each other. The suggested scheme protected from security attacks. In addition, the proposed scheme provides better protection and better functionality characteristics in terms of connectivity and processing costs but it still does not address certain types of attack.

Nikooghadam et al. [23] presented a scheme for developing a stable authentication based on an elliptical curve for drones to ensure surveillance of a smart city. The scheme demonstrates supporting security criteria and resisting established attacks, while incurring low computational and communication costs.

Deebak and Al-Turjman [65] suggested a scheme of IoD to gather sensible knowledge independently. The scheme is implemented to reduce the computation expense of the authentication protocol. The suggested scheme is constructed to provide a valid authentication time for robustness between the IoT devices. However, the proposed scheme has not addressed privacy leakage. Bera et al. [22] discussed a scheme to secure a data management system between entities of IoD communication based on blockchain. The proposed scheme has the ability to resist multiple possible attacks. In addition, comparative analysis shows that the proposed scheme offers better conditions for security and reliability and also has fewer overheads for computational and communication compared to similar schemes. However, the proposed scheme required more investigation of blockchain technologies to support deployment in a real IoD environment.

Wang et al. [66] presented a scheme of joint learning segmentation combined with a model of conditional random-field and designed model based-enhanced U-net to improve the accuracy of object segmentation and to solve the problem of inappropriate edge recognition. The findings of the presented scheme indicated that the precision of the segmentation of the ground object increased up to 86.1%, which is a promising development.

Tian et al. [2] introduced a framework for authentication to protect privacy. The framework guarantees the efficiency of authentication when deployed on resource-constrained small-scale UAVs using the lightweight online/offline signature design. The presented framework is enabled to predict authentication by exploring mobile edge computing (MEC) in view of the high mobility of UAVs to further reduce the cost of authentication for future authentication activities. Moreover, the framework allowed privacy security in terms of the UAV identity, location, and flying routes by developing a username buffer and updated strategy of the public key. However, the study is missing the use of formal analysis to compare current authentication schemes in detail.

Ever [3] designed a framework for UAV environment authentication using elliptic-curve crypto-systems. The proposed framework was tested to ensure that it is immune to substantial well-known possible data confidentiality, shared authentication, password guessing, and key impersonation related attacks. However, the study has not covered privacy issues.

Choudhary et al. [25] proposed a framework for a security based-neural network, which implements MAC protocol managed by the length of Macaulay. The proposed framework enhances the sufficient duration of comprehensive control over ties to enhance the security aspects of IoD and introduces countermeasures against identified cyberattacks. However, the proposed framework does not support the combination of a security protocol with channel authorization and real-time performance.

Nouacer et al. [67] designed a framework to secure the architectures based drone for software and hardware. The framework holds a holistically built ecosystem ranging from electronic components to applications. Moreover, the framework enabled a tightly integrated solution for multi-vendor and scalable drone embedded architecture and an appropriate tool-chain. The problems of latency and computing capacity, on the other hand, are not fully discussed.

Sharma et al. [68] suggested a protocol that depends on congestion, which offers both symmetric reliability and congestion control based on rate adjustment. The protocol aims to minimize packet drops, maintain fairness, and boost energy quality. In addition, the protocol offers greater energy effectiveness and efficiency. However, the proposed protocol does not cover security and privacy issues.

Mukherjee et al. [54] presented a mechanism to enhance protocol solution as an amalgam with the opportunistic routing mechanism.

The mechanism kept monitoring the history of the encounter and the transitivity of encounter time. The analysis of the improved mechanism has been further checked and a substantially improved performance in terms of publisher bandwidth, latency, and run time has been observed. The findings of the proposed mechanism showed efficiency improvement for both memory usage and energy dissipation. However, although the problems of latency and computing power have been mitigated, there are concerns of bandwidth that remain unresolved.

Bera et al. [69] suggested a mechanism of access control for unauthorized UAV detection and mitigation. The proposed mechanism is enabled to protect data from a drone (UAV) to the server of the ground station and the abnormal data for the identification of unauthorized UAVs. The results of the proposed mechanism illustrated the possibility to conduct big data analytics on authenticated transactional data recorded on the blockchain. However, the study is missing the cover privacy and attack issues during the transfer of data form the UAV to the server.

Weng et al. [70] suggested a method to improve mobility compensation of drone based IoT. The method involves phases of frequency offset approximation and relative velocity measurement. The proposed method shows beneficial gains of mobility compensation of drones via the simulations of Monte Carlo, but the study has not considered security and privacy issues of drones’ mobility.

Dawaliby et al. [71] designed an architecture-based blockchain to handle IoT drone operations while retaining and security. The results of the introduced architecture obtained in the case of practical agricultural usage illustrate the usefulness in reducing signaling and operating time, improving the percentage of effective maintenance operations, and providing trust and protection in autonomous drone management. The study needs to develop a platform to better meet the demands of IoT devices and use different placement strategies.

Gallego-Madrid et al. [72] introduced an architecture based on the ability of multi-access edge-computing (MEC) to host a drone-board virtualization platform to provide an intermediate processing layer running virtualized networking functions (VNF). The results of the presented architecture have shown significant communication improvements using LoRa-drone gateways in terms of link availability and covered areas, particularly in vast controlled extensions or at points with difficult access, such as rough zones. The study needs to conceptually expand the architecture with software-defined networking and artificial intelligence to dynamically pre-configure the networks according to the current context.

Koubâa et al. [28] proposed an architecture-based cloud for drone management to control, monitor, and communicate over the Internet. The architecture allowed smooth contact with the drones over the Internet, enabling them to be operated anywhere and at any time without distance limitations. In addition, the architecture provided drones with access to cloud computing services to offload heavy computations. Furthermore, the proposed architecture demonstrated the efficacy performance assessment analysis using a real drone for a real-time monitoring application. However, the study has not covered privacy issues of data movement between drones and clouds.

Rehman et al. [73] suggested an algorithm to support information searching that exploits the ranking of services in a drone network. The results of the proposed algorithm have shown that, with its average execution time not exceeding 2.5 ms on two separate machines, the proposed knowledge searching algorithm proved to be successful. The study has not defined in detail the services, classes, and mathematical models to rating parameter, in order to offer better optimization design and performance.

Chang et al. [74] presented an algorithm to improve cost estimation for multiple in-flight rerouting. The experimental flight test of the proposed algorithm was conducted ten times on different routes. In addition, the findings indicate that the adjusted cost estimation algorithm has a rerouting cost estimate of more than 92% accuracy under multiple rerouting path-points. However, the study did not use AI and IoT to develop the autonomous flight algorithm’s foundation.

Huang et al. [75] introduced an algorithm to support the analysis of the computational complexity of IoD. They demonstrated the efficacy of the proposed algorithm, detailed computer simulations are carried out, and a comparison with identified objectives is given to determine performance gains. This study did not disprove assumption and identify potential locations for drone-BSs to provide service to mobile users. Another flaw is that environmental variables such as rain and wind were not taken into account in this study.

Yao et al. [76] presented an algorithm to minimize the energy consumption of the drone. The suggested algorithm obtained the optimal solution and transformed the convex optimization problem, and then updated the Lagrangian parameters with an updated Newton method. The results of the simulation showed that the proposed algorithm performs better than the current algorithms. However, the issues of latency and computational power are not completely addressed.

In conclusion, there have been few recent studies that have made advances to contribute to the research fields. For example, only a few studies addressed schemes, frameworks, architectures, and mechanisms, as well as systems for enhancing and improving the security and privacy of IoD services. Therefore, further research and investigation of issues are required for IoD environments.

## 7. Challenges and Future Research Trends

This section highlights the most crucial challenges and future research trends on the Internet of drones.

### 7.1. Privacy and Security-Related Challenges

Since government agencies grant licenses for drone usage in civilian and commercial applications, the modern IoD is equipped with numerous connected sensor devices. These devices are vulnerable to various threats including hijack, human error, and loss. These issues should be given higher priority in the design of drone applications.

Integrating AI-based security detection with blockchain to protect the devices and a more sophisticated defense system to control and monitor the application will help reduce the security threat to IoD [77]. In fact, when referring to UAVs one may have to stress the “temporary” relation since these devices could be “untethered” from land. Communications networks are vulnerable to jamming attacks that could catch new vehicles or shut down those vehicles by manipulating their controls. To ensure this goal is met, high integrity protected data links must be used between the flight and ground controllers, and the Federal Aviation Authority (FAA). Technological advancement such as augmented telecommunication and data protection technologies can help avoid the security breaches of the future. Nevertheless, anti-jamming using high powered signal encoding is possible with modern high technology. Communications protection depends on the communication channel frequency, communications media, technology, and the relationships between each other. Usually, encryption algorithms that operate at low bit rates are a bigger problem than those that run at high bit rates as there are more costly categories. Therefore, lowering costs are often followed by a reduction in security or in the number of operations [78].

Additionally, owing to the UAVs autonomous nature, safety is a major concern. The UAVs are noisy and not easy to monitor, hence more safeguards should be established to maintain data protection. There will be many possibilities, implications, and events from the analysis of UAV. To secure UAV systems, using many techniques including redundant sensory devices and security systems that are linked to the UAV ground station and others, should be addressed. Other methods of preventing UAVs from being hacked have been reported. The recommendations from the defense department (DoD) discuss taking steps to counter hijacking. Thus, there are very few details on anti-terror strategies for coping with hijacking. The potential danger of hacking in the air can look as malware in the software created, spoofing attacks on communications-relay, and manipulation hardware or software. One of the possibilities to counter the attack is for the control station to manipulate the electromagnetic field of the carrier signal with hardware on the malicious attacker side. Other forms of attacks include stealing data and gaining control of device functions. To minimize the implications of these conditions, enabling the use of risk reduction standards that could be exercised in court is essential. Which risks are compatible to achieve the highest utility? Based on the available details, these decisions will lead to potential choices to be made. A ground radar device that detects vehicle hijacking is potentially useful by telling the driver that the vehicle has been electronically changed. The only drawback will be the expense of developing, equipping, evaluating, and implementing the product since they had a dollar for dollar return on investment [79,80].

### 7.2. Global Resource Management-Related Challenges

Resource distribution is critical in serving productivity and reducing cost and can be divided into two groups, the global allocation of resources and local allocation. Global concerns concentrate on the expenditure spent on the global resources of time, energy, and equipment. In addition, the maximum global productivity can be deployed by various equipment in the Internet of everything (IoE), such as edge computers, cloud servers, and UAVs. Under digital media transmission in IoE, global efficiency can be increased by networking algorithms and video coding. Energy and power are allocated for every role based on committed performance. For instance, in a smart building scenario (such as in a local network), an efficient data collection algorithm is good for increasing the user’s data rate. In addition, for the flying terminals, the energy management system is essential for making use of limited energy. This is the concept of allocating resources at various nodes [81].

In fact, a challenge for UAV applications is to preserve efficiency while facing cost issues. First, developing protocols for the provisioning of network services based on information rate and requirements of computation is a challenge. The problems to be addressed are for systems that are clustered, use of cloud resources, and autonomous services. In addition, the most popular economic activity is “resource allocation” since it includes competing interests of individuals and companies or services to receive particular resources. The next challenge is discovering reliable and cost-effective ways to assign consumer demands. Another difficulty in resource utilization centers on determining how best to specify the priorities and paths to the minimization of activity energy expenditure. Moreover, resource mapping aims to offer an equal amount of services to buyers and suppliers. Resource mapping may face problems such as mapping the physical components and the rational distribution of resources to fulfill the requirement, which is often hindered by physical limitations and obstacles. Designing algorithms that can use generic methods to quickly obtain a mapping process fast is critical. The application’s compatibility with the UAV platform needs to be checked by installing corresponding hardware prerequisites. Yet another difficulty to be resolved would be to build models that can correctly measure the performance of multi-core CPUs, PC storage medium, communication channels, and information center data stores. In addition, this consideration may be a problem when mapping resources [79].

### 7.3. Sensor Communication-Related Challenges

The IoD sensor’s communication protocols are designed with lightweight and highly sensible objects which have a substantial chance for data loss or receive the wrong data from other nodes. Furthermore, sensors face routing issues when communicating with multiple drones with a network center. Some of the drone manufacturers use cheap hardware components which increase the chance of other network communication issues namely, high throughput, latency, and delay between the device and center [82]. To address these issues, next-generation network technologies such as 5G, intelligent routing, narrowband-Internet of things, LoRa-based IoT systems, Sigfox, NB-IoT, and LTE-M need to support connection choices in the future. Lastly, the standardized policy should be developed to use the authorized component for drones’ communication protocol.

### 7.4. Coordination and Tasks Scheduling-Related Challenges

Cloud computing and edge computing are going to be combined in the future to handle computationally intensive IoT applications. These intelligent external intelligent network applications need an appropriate centralized AI analysis and individual big data analysis for coordination. Regarding the situation-target scheduling, the applied intelligence first examines the gathered information on the necessary computing tasks and then decides whether the tasks are required to be submitted to the remote cloud. The object of migration is different from the requirement of resources for a real-world data or application. Computer architecture and networking will be adapted to international performance. This method can be changed at any time for possible changes and improvements. This will certainly become a regular theme [83,84,85].

### 7.5. Drones Distribution and Deployment-Related Challenges

Apart from data confidentiality, data sharing and access control are challenges that face IoD deployment. For instance, in the application where a set of drones can collaborate to collect road traffic data of different regions, how to securely and efficiently share these collected data (e.g., in the sense that only authorized entities have access to the data) remain an ongoing challenge [20,22].

## 8. Conclusions

Unmanned aerial vehicles or drones are helpful in raising efficiency in everyday life. With the number of drones in low altitude air territory, linking drones into the Internet of drones (IoDs) will help enhance the safety and performance of the drone aircraft. However, privacy, security, and communications are still a big concern of the governments. We have addressed key prospects for the association of IoD. This article provides a taxonomy of the IoD based on the most important perspectives. We have included commercial cases that are the most common and illustrate the most recent techniques used for IoD. Finally, we have addressed the obstacles IoD faces and provide some direction for future studies.

## Figures and Tables

**Figure 1 sensors-21-05718-f001:**
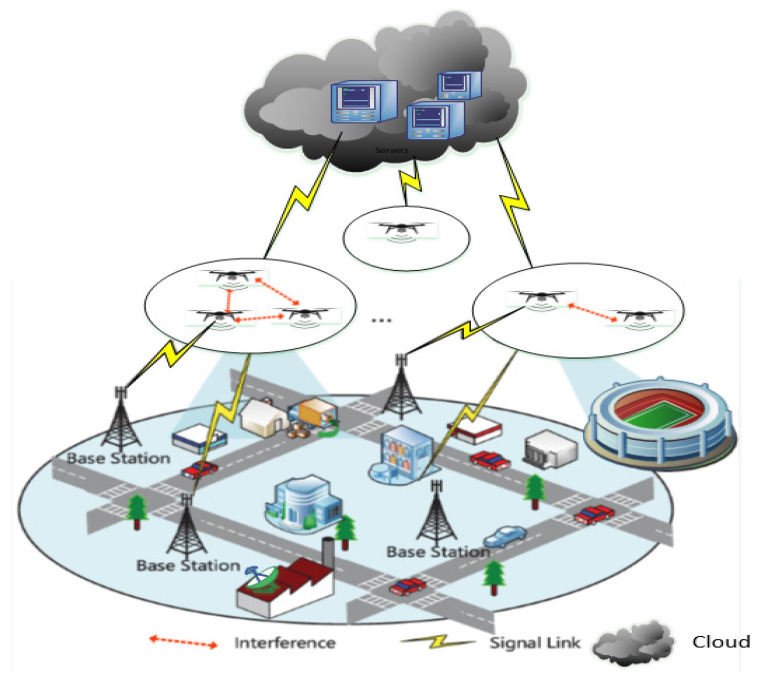
IoD environment.

**Figure 2 sensors-21-05718-f002:**
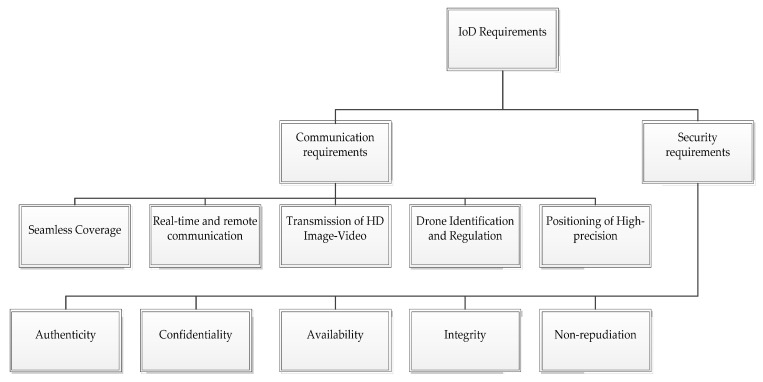
The key requirements of IoD.

**Figure 3 sensors-21-05718-f003:**
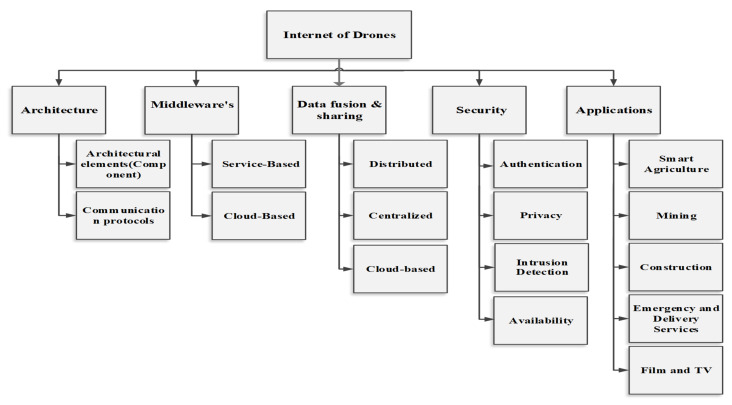
Taxonomy of IoD.

**Table 1 sensors-21-05718-t001:** Related works.

Survey Paper	Year of Publication	Highlight	Comparison of Current Research Works in Terms of	Domain
This work	2021	Comprehensive study covering requirements, taxonomy, recent advances, and challenges of future research trends	Drone applications Commercial case studies Recent advances in IoD	IoD
Boccadoro et al. [8]	2021	Drone adoption for the economy and society	Drone applications	IoD
Rejeb et al. [9]	2021	Humanitarian drones	The humanitarian drone’s capabilities and performance	Drones
Yahuza et al. [10]	2021	Evaluate IoD network security and privacy	Drones classification	IoD
Ayamga et al. [11]	2021	Review the advancements in agricultural, military, and medical drones	No comparison table	Drones
Merkert and Bushell et al. [12]	2020	Need for more policy and management response to drone usage	Countries and regions contributing	Airborne drones
Zaidi et al. [13]	2020	State of the art of IoFT	IoT, FANET, and IoFT MANET, VANET, UANET, and FANET UAV system, multi-UAV system, and FANET	IoFT
Yaacoub et al. [14]	2020	Analysis to assist ethical hackers in recognizing the current vulnerabilities of UAVs	Frequencies of drones communication Drones cyber-attacks Analytical drone review	Unmanned Aerial Vehicles
Al-Turjman et al. [15]	2020	Review the performance assessment techniques and cybersecurity applications	Flight stacks Objective, constraint, area, and assessment approach Mass simulators Testbeds	IoT networks
Alsamhi et al. [16]	2019	Improve smart cities’ real-time implementation	Surveys and current work Drone and IoT for smart cities	Smart cities
Fotouhi et al. [7]	2019	Overview of advancements supporting compatibility of UAVs into mobile networks	Characteristics of drones Comparison of aerial placement	UAV Cellular
Wazid et al. [4]	2018	Define the strengths and limitations of user authentication of IoD	Authentication schemes Communication overheads Security features and functionality	IoD
Bagloee et al. [17]	2016	Vehicles’ connection and infrastructure of future growth of AV	No comparison table	Autonomous vehicles
Gharibi et al. [1]	2016	Investigate and extract key concepts of networks	No comparison table	IoD

**Table 3 sensors-21-05718-t003:** Commercial case studies.

Case Study	Middleware Strategy	Data Sharing	Supported Applications and Services
Matternet	Cloud-based	Cloud-based	Healthcare and logistics
Aria Insights	Service-based	Distributed	Oil and Gas
DroneSmartX	Cloud-based	Cloud-based	Agriculture Energy Manufacturing Oil and Gas
H3 Dynamics	Cloud-based	Cloud-based	Real Estate Mining Oil and Gas

**Table 4 sensors-21-05718-t004:** Recent advancements in IoD.

References	Architectural Organization	Middleware Strategies	Solution Type	Proposed Solution	Supported Applications and Services
Almulhem [63]	Communication protocol	Cloud-based	Model	Provide a holistic view of threats affecting an IoD scheme	IoD system
Zhang et al. [64]	Component	Service-based	Scheme	Attain security and withstand various attacks	IoD architecture
Nikooghadam et al. [23]	Communication protocol	Service-based	Scheme	Develop authentication for drones to ensure surveillance of smart city	Smart City Surveillance
Deebak and Al-Turjman [65]	Communication protocol	Service-based	Scheme	Reduce the computation expense of the authentication protocol	IoD infrastructure
Bera et al. [22]	Communication protocol	Service-based	Scheme	Secure data management system between entities of IoD communication	IoD environment
Wang et al. [66]	Communication protocol	Service-based	Scheme	Improve the accuracy of object segmentation	IoD
Tian [2]	Communication protocol	Service-based	Framework	Secure authentication to protect privacy of UAV	IoD communication
Ever [3]	Communication protocol	Service-based	Framework	Secure authentication of UAV environment	IoD applications
Choudhary et al. [25]	Communication protocol	Service-based	Framework	Enhance IoD security aspects	Military IoD
Nouacer et al. [67]	Component	Cloud-based	Framework	Secure architectures based drone for software and hardware	IoD architecture
Sharma et al. [68]	Communication protocol	Service-based	Protocol	Minimize packet drops, to maintain fairness and boost energy quality	Wireless Sensor Network
Mukherjee et al. [54]	Communication protocol	Cloud-based	Mechanism	Enhance protocol	EdgeDrone
Bera et al. [69]	Communication protocol	Cloud-based	Mechanism	Access control for unauthorized UAV	IoD environment
Weng et al. [70]	Communication protocol	Service-based	Method	Improve mobility compensation of drones	IoT drone
Dawaliby et al. [71]	Component	Cloud-based	Architecture	Handle IoT drone operations	IoT drone
Gallego-Madrid et al. [72]	Component	Cloud-based	Architecture	Provide an intermediate processing layer running (VNF)	Drone Gateways
Koubâa et al. [28]	Communication protocol	Cloud-based	Architecture	Control, monitor, and communicate over Internet with drones	IoD
Rehman et al. [73]	Communication protocol	Service-based	Algorithm	Enable information search to explore the ranking of services	Drone network
Chang et al. [74]	Communication protocol	Service-based	Algorithm	Improve cost estimation for multiple in-flight rerouting	IoD
Huang et al. [75]	Communication protocol	Cloud-based	Algorithm	Support analysis of computational complexity	IoD
Yao et al. [76]	Communication protocol	Service-based	Algorithm	Minimize the energy consumption of the drone	IoD

## Data Availability

Not applicable.

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
