# Peer review of "The Internet of Drones: Requirements, Taxonomy, Recent Advances, and Challenges of Research Trends"

_sensors, 2021, doi:10.3390/s21175718_

Round 1

Reviewer 1 Report

The corrections introduced to the paper by the Authors are satisfactory. 

Author Response

Thanks much 

Reviewer 2 Report

The manuscript entitled “The Internet of Drones: Requirements, taxonomy, recent advances and challenges of research trends” is an interesting contribution that aims at discussing a subset of the most relevant aspects of the Internet of Drones.

Despite the authors' work, the manuscript still deserves the authors’ attention, and it is the Reviewer’s opinion that the work should be further reviewed before it can be taken into consideration for publication. In what follows, a detailed explanation of the reviewer’s decision is given.

Overall opinion:

The Reviewer would like to suggest the authors read and study the following reference

Boccadoro, Pietro, Domenico Striccoli, and Luigi Alfredo Grieco. "An extensive survey on the Internet of Drones." Ad Hoc Networks (2021): 102600. https://www.sciencedirect.com/science/article/abs/pii/S1570870521001335?dgcid=rss_sd_all

This is motivated by several aspects. First of all, there’s general confusion on the meaning of “Internet of Drones”. This does not mean that “things” in the Internet of Things paradigm are simply substituted by “drones”. To better frame this, the authors are encouraged to study their reference number 1.
Second, all the references to “communication protocols” and or “communication paradigm” and or “architecture” are misleading. This is also clarified by subsection 4.1.2, in which the cited “Communication Protocols” are not communication protocols. Indeed, MAVLink is proprietary communication technology and ROSLink is an extension of the ROS operating system. Indeed, the Internet of Drones is manyfold and composed of several communication protocols at different levels. Moreover, the IoD is a networking architecture that includes multiple communication technologies depending on the link, the expected Quality of Service, and the specific service level.
Third, Table 4 should summarize what was discovered but the general confusion does not help in understanding what a “communication protocol” is. Which protocol? At which level of the protocol stack? In which application? Which are the other protocols for all the other levels of the stack? And lastly, which are the involved protocol stacks?
Forth: drones are not necessarily “small” UAVs, and the IoD includes and integrates all “drones”, regardless of their being fixed-wing or rotary, for example. Before writing a survey paper on this theme, this must be understood in one with all the relevant technological aspects.

In table 4, what is the difference between “IoD system” and “IoD Architecture”?

It is not clear why some sentences are reported in red. Is this a revised version of a paper? If so, a rebuttal should be given in one with the previously submitted version of the manuscript.

Overall, it must be said that this paper is all about listing some contributions. Unfortunately, no critical analysis is given, no relationship is established between the references. Moreover, this work does not critically analyze the investigated state of the art. As a consequence, section 7 is almost full of well-known concepts.

Detailed comments:

In Section 4:

It is not clear if the links reported on page 12 are references or not. In case, they should be added to the References.

In Section 7:

Subsections 7.3 and 7.4 must be revised because of their technical flaws.

NarrowBand-Internet of Things is shortened as “NB-IoT”, as well as “LoRaWAN” is the proper way to refer to the second layer of the LoRa-based IoT systems.

As for the references,

The authors should consider enriching the related state of the art to include more.

Reporting authors’ list with “et al.” should be done anytime authors are more than 3. This rule should be used for all the included citations.

Author Response

Thanks much for your useful comments 

Round 2

Reviewer 2 Report

The authors addressed all the comments.

This manuscript is a resubmission of an earlier submission. The following is a list of the peer review reports and author responses from that submission.

Round 1

Reviewer 1 Report

The paper presents a survey of the IoD (Internet of Drones). It covers aspects of applications, communication requirements, security, and a listing of aspects to consider (such as middleware, data fusion, etc.). 

Though it would be interesting to have a well-organized survey on a specific aspect of the IoD, the paper is not convincing due to the following major weaknesses:

  • The paper lacks a clear focus. The paper superficially targets several aspects that are (also) important in networks of drones, yet neither are the aspects (such as networks/communication) discussed at the required level of detail. 
  • When speaking about the IoD, I expected more insights into current networking technologies used in drone networks - this is totally missing.
  • The mentioned aspects seem not to target the very specific characteristics of UAVs, e.g., movement in 3D space, powered mainly by battery, thus come with a short life-time. It is unclear in which way the requirements differ from any other networked embedded system.

Detailed comments:

  • Editorial: English proof reading is required (articles and typos).
  • There is a related survey that seem not to be included:
Azade Fotouhi, Haoran Qiang, Ming Ding, Mahbub Hassan, Lorenzo Galati Giordano, Adrian Garcia-Rodriguez, and Jinhong Yuan. 2019. Survey on UAV Cellular Communications: Practical Aspects, Standardization Advancements, Regulation, and Security Challenges. Commun. Surveys Tuts. 21, 4 (Fourthquarter 2019), 3417–3442. DOI:https://doi.org/10.1109/COMST.2019.2906228
  • Figure 2: Key future requirements is strange to mention, as there may be future communication and future security requirements as well. I recommend to rework this figure.
  • page 7 (and following): Subsections consisting of title and one sentence makes this part of the paper look more like a technical manual than a review/survey paper and is not well readable.
  • Table 2 gives a good overview of possible application fields.
  • Later in the paper (Section 6 and following), suddenly the IoT is discussed and focused on (i.e., the Internet of Things). So, the question is how is the IoD different from the IoT or, on the other hand, which technologies are used in both Io* fields? This should be discussed and clarified.
  • References: Some references do not specify the publication venue (e.g. reference 26) - this should be corrected.

Reviewer 2 Report

The subject of the article is interesting, however, after reading the article, the following issues were formulated. 

- lines 255-266: improper structure of the content - it is not known why each "Security requirement" is marked with another sub-item, ie 3.2.1 - 3.2.6, since these are only single sentences? 

- lines 270-307: improper structure of the content - it is not known why each "Key future requirement" is marked with another item, ie 3.3.1 - 3.3.13, since these are only single sentences? 

Basic remarks: The title of the article is too broad, while the content of the article is a cursory characteristic of the IoD issue. It is difficult to consider this article for a scientific journal. This is an article that popularizes science - it is for a wide range of average readers rather than scientists.

Reviewer 3 Report

Please find the constructive review comments below:

  1. In Abstract,
    • Rewriting/Rephrasing the abstract shall be considered for better interpretation of the survey made clearly.
  1. In Introduction Section,
    • Line 31 – “The key advantage of a UAV with fixed wings is that it is much easier than a UAV with rotary wings.” Is the statement context missing here?
    • Figure 1 shall be Updated. Interference indicators are not visible. What do the green dotted lines indicate?
  1. In Requirements of IoD Section,
    • In 3.1.1, Text content is repeated from Line 194 to 204 (Same as from line 177 to 193) Copy Paste Error?
    • In 3.2, Rather than just specifying the list of security requirements, apart from reference [4], cite other considerable methods/techniques/approaches if any to ensure how these security requirements are addressed in IoD.
  1. Opportunities
    • The opportunities sections seem to be redundant.
    • How is it only limited to Disaster management, Cloud and Smartphones, and Public safety?
    • Merging this section with the applications section shall be considered as it also includes “Emergency and Delivery Services.”
  1. Taxonomy of IoD
    • Substantiate the architectural key elements and protocols of IoD (apart from key elements of IoT) with proper citations.
    • Tabulating the communication protocols used in IoD and the protocol selection/usage criteria.
  1. In Applications section
    • In table 2, Check the relevance of [30] and [31] citation. A more relevant work shall be cited respectively for construction and Emergency service applications.
  1. Challenges and future research trends section
    • Incomplete statement in section 8.4 – “To address these issues next-generation network technologies like 5G, intelligent routing, 876 NbIoT, Lorawan, Sigfox, nb-iot, and lte-m. ”(line 875-876)
    • Citation for section 8.5 shall be added.

Text/Formatting Corrections (To List few..)

  1. Check for text format from line 159 to 167. (Font Style, Spacing)
  2. Check for text format from line 255 to 307. (Text alignment)
  3. Line 320, “A certain situation can occur when (While?) acquiring large-scale sensor data.”
  4. Table 2, Maintain consistency in the reference column. (Citations followed by links or Links followed by citations)
  5. Check line 553 and 554 – “A broad variety of Internet Things applications in the fields, mining and industrial 553 automation services are being opened up by drones acting as sensor devices”
  6. Line 629, “Recent Advances” shall be made as “Recent Advancements”

General Comment

  1. Adding/Using infographic images wherever appropriate shall be added to improve the readability of the survey paper.